# Economic Complexity and Human Development: Moderated by Logistics and International Migration

**Emilie Sophie Le Caous [1,\*] and Fenghueih Huarng [2]**

1    College of Business, Southern Taiwan University of Science and Technology (STUST), No.1, Nantai Street, Yongkang District, Tainan 71005, Taiwan
2    Department of Business Administration, Southern Taiwan University of Science and Technology, No.1, Nantai Street, Yongkang District, Tainan 71005, Taiwan; fhhuarng@stust.edu.tw
\*    Correspondence: emilielc@stust.edu.tw; Tel.: +886-983691768

**Abstract:** Living in a world where we can expand our economic wealth and the richness of human life is the core of the human development concept. Greater well-being for all can be achieved by improving people's capabilities and more importantly, by giving individuals the ability to use their knowledge and skills. The economic complexity index (i.e., ECI) is a new indicator that defines a country's complexity. Through a vast network, citizens can transfer an enormous quantity of relevant knowledge, leading to the creation of diversified and complex products. However, the relationship between economic complexity and human development is not that simple. Thus, this paper aimed to understand it deeper—international migration and logistics performance are used as moderators. Hierarchical linear modeling was the statistical tool used to analyze two groups of countries from 1990 to 2017. For robustness and to deal with possible endogeneity issues, different year lags were also included. The results show that international migration and logistics performance are decisive moderators as they change the relationship between economic complexity and human development.

**Keywords:** economic complexity; human development; logistics performance; international migration; gender inequality; social development; hierarchical linear modeling; HLM

## 1. Introduction

Development is one of the most complex but fundamental terms in our lives. For a long time, GDP per capita was considered the primary indicator of countries' wealth. Nowadays, the focus includes the economic point of view, but not only. Countries worldwide should provide an environment where their citizens can expand their knowledge and skills to access more opportunities, expand their choices, and satisfy their basic human needs to reach a higher level of well-being. Better living conditions in terms of education, health and income, are affected by many factors, such as governance, social development, and inequalities [1–4]. A country should provide a political environment where individuals feel safe and free to share their views, where citizens believe in their government and can become civic participants, where young generations have great opportunities for jobs, where gender gaps are reduced, and where the minorities are included.

We believe that economic complexity could help to understand the level of human development. By diversifying their productive structure and becoming economically complex, countries can reach high economic growth levels [5,6]. To measure the productive structure, we used the economic complexity index (i.e., ECI). The ECI is "the indicator of the composition of a country's productive outputs and the structures that emerge to hold and combine knowledge" [6]. The economic complexity index can quantify the knowledge and capabilities available in a country. By exchanging the knowledge acquired through education and work experiences, individuals create a vast network, where the collective knowledge is transferred and improved, allowing the creation of diversified and complex products. As a country innovates and becomes economically complex, individuals

should experience greater freedom regarding choice, capabilities, and life satisfaction [7–9]. Therefore, a country should have a broad set of skills to produce diversified and complex products and services. Unfortunately, not every country possesses the right number of skilled workers. The migratory attraction has been used to save time and money to face this issue and more importantly, to access human potential formation [10]. Globalization has facilitated the flux of migration, and the effects could be either positive or negative. Moreover, due to globalization, logistics become primordial as it is the backbone of international trade that interlinks the global value chains [11]. A performant logistics system makes the multi-model transport and distribution more efficient, facilitates trade, and improve sustainability.

This paper aimed to analyze the relationship between economic complexity and human development for low human development index (HDI) countries and high HDI countries. We believe that international migration and logistics performance may play as moderators and influence the relationship between economic complexity and human development. To analyze the yearly and country effects of the different predictors, we use hierarchical linear modeling (i.e., HLM) as our statistical tool. Our dataset includes two groups of countries with available data from 1990 to 2017: 59 high HDI nations and 55 low HDI nations. Finally, different year lags are analyzed to achieve robustness and deal with possible endogeneity issues. Since, in the last decade, there has been a growing interest in the academic field on economic complexity and its linkage to sustainability [5,12], we contribute to the field of research by analyzing the moderating effects of international migration, and logistics performance which is an underestimated indicator that plays a key role in explaining the relationship between economic complexity and human development. This paper is structured as follows: in the next section, we discuss the relationship between human development, social development, economic complexity, international migration, and logistics performance. Then, we present our data and the HLM methodology. In Section 4, we analyze and interpret the statistical results. Finally, we discuss our findings and the possibilities of future research.

## 2. Literature Review

### 2.1. Comprehending Human Development

Giving individuals the tools to enlarge their opportunities to reach a higher level of well-being is the key to human development [1]. The human development concept focuses not only on the economy and the balance of welfare conditions among citizens but also on the process that allows individuals to satisfy their basic human needs [13,14]. Conceptualizing such a concept is not that easy. Through the years, many indexes have been created, such as the human development index (HDI), the inclusive wealth index (IWI), the adjusted net savings (ANS), the genuine progress indicator (GPI), the inequality adjusted human development index (IHDI), the better life index, and recently, the planetary pressures adjusted human development index (PHDI) [15–18]. In this paper, we used the HDI to measure human development. We do not want to use other indexes that include environmental or social characteristics, as we believe that they may play as mediators or moderators that could explain the relationship between economic complexity and human development. The HDI is "a statistical tool used to measure countries' overall achievements in its social and economic dimensions" [19]. Three keys dimensions compose the HDI: health, education, and income [20] (see Appendix B).

The HDI mainly focuses on individual capabilities such as knowledge and health. However, essential elements that can improve individuals' quality of life, such as the basic prerequisites for human security and survival, political rights and freedom, and social cohesion, are not included in the HDI [21,22]. Social development is primordial as it results in collective actions, social accountability, and inclusion, leading to people's empowerment [23]. Social development includes different dimensions—interpersonal safety and trust, the inclusion of social and ethnic minorities, gender equity, social cohesion, civic engagement, and community ties [23]. Gender inequality can be considered as a social

and cultural indicator. Women's participation in the labor market has a decisive impact socially, economically, and culturally, leading to changes in our societies [24]. As better equality among genders in education, employment, and income distribution increases the level of human development, high-income nations usually focus on gender equality [25] and try their best to reduce inequalities between genders. Even though improvement has been made among low-income countries, there are still gaps between men and women in terms of education, job rewards, and deprivations [26]. If a country can protect the women's role and active participation in society, literacy and education will increase, the growing population will be controlled, and health will improve [24].

If a country is socially sustainable, inequalities and exclusion are reduced [27]. Therefore, in terms of job opportunities, the allocation of benefits and social resources against vulnerable groups in society, discrimination reduction should be a priority to governments. Discrimination is a critical issue leading to social exclusion and is not easy to measure. One way is to ask individuals if they feel that they have been mistreated based on their identity. Immigrants and members of ethnic minorities usually feel discriminated against based on their "ethnic or immigrant origin, gender, age, disability, sexual orientation, religion or belief" [28,29]. If an individual cannot "participate in social, economic, political and cultural life" [29], they are considered socially excluded. Therefore, achieving well-being is difficult as they may not have job opportunities, access to proper housing, or even lack access to education and healthcare services. If a country can include minorities, the economy will become more efficient due to the human resource potential [30]. It is important to note that social exclusion is not only related to poverty but also disability and age. Aging is important as it is linked to our body and societal issues such as politics, sociology, civil participation, and public healthcare. It impacts both low- and high-income countries. Aging is directly related to life expectancy: with healthcare systems and medicine, the world population has been living longer [31]. Other reasons for the decrease in mortality rates are better lifestyles and ameliorations in the work environment. Aging has also impacted fertility [32]. As people live longer, they tend to have children later in life, which is also explained by contraception improvement. Moreover, older individuals are usually active community members who follow the law, decreasing the level of criminal rates.

Therefore, civic activism is another critical element of social development, defined as "the social norms, organizations, and practices that facilitate greater citizen involvement in public policies and decisions" [30]. According to the 2030 Agenda of Sustainable Development, civic engagement is key to the social development of a country as it allows individuals to participate in the media, impacts policies, holds public authorities accountable, and changes behaviors and cultural norms. Civic activism is a crucial characteristic of human development [33]. Individuals that participate in the development of their civil and institutional society impact social justice and freedoms. Engagement in society is significant for young people, as it is at this stage that they experience the roles attributed by society, such as voting and finding a job [33]. Unfortunately, with COVID-19, young people are among the first ones to lose their jobs. Youth unemployment has a long-term negative impact on one's employment career, opportunities of jobs, and wage levels [34]. Youth employment is a policy challenge and a key determinant of human development for low- and high-income countries. Since the 1990s, the global labor force participation has decreased, displaying greater retirement opportunities and higher life expectancy [35].

Every individual wants to live in a society that is peaceful, free, and prosperous. Therefore, countries should possess a social system improving cultural values and better governance [36]. Governance is a process through which formal and informal institutions interact and decide which policies should be developed and implemented in order for a country to shape the rules necessary for the application of the authority [37,38]. Governance is vital for many reasons, the most important being freedom, which allows creating a bundle of opportunities [38]. According to Sen, development eliminates many types of unfreedoms and enables individuals to use "their reasoned agency" [39]. Governance provides security for citizens and facilitates prosperity and equity. The Sustainable Devel-

opment Goal 16 (SDG 16) promotes "peace, justice, and strong institutions" [40], linking every society that wishes to reach these goals in environmentally sustainable ways to their governance. A solid understanding of governance is of great importance as institutions' quality has a strong influence on nations' development [41], but more importantly, it facilitates the introduction of effective policies and thus allows the accomplishment of all the other SDGs [38]. Many dimensions have been proposed and used to measure governance [42–44]. The World Bank developed the world governance indicators, namely the control of corruption, voice accountability, political stability and the absence of violence, the rule of law, regulatory quality, and governance effectiveness. According to Kaufmann and his collaborators [45], the world governance indicators (WGIs) provide information regarding a country's aptitude to elect, supervise, and replace authorities; develop and implement policy initiates; and the extent to which citizens and political dignitary respect the governing institutions. In this research, we use the rule of law and regulatory quality. Regulatory quality focuses on the perception of market-unfriendly policies and the burden of excessive regulations and their impact on trade. Rule of law is "the extent to which agents have confidence in and abide by the rules of society" [45].

## 2.2. Economic Complexity and Human Development

According to the concept of creative destruction, innovation is the key to drive the economy [46]. Technological change usually comes with the transformation of the economic structure and societal change. Incumbent firms should be able to adapt to the new technologies in order to stay competitive in the market. Knowledge is primordial as it is the source of people's innovative ideas [47]. According to Arrow [48], knowledge is a source of increased returns of scale. For Romer [49], knowledge is an asset in the productive process as it is a non-scarce resource. The accumulation of knowledge allows long-term growth. Knowledge is a nonrival good that can be used at the same time in other activities. Therefore, technological goods are nonrival and excludable, which means that the company who innovated can ask other companies to pay for its innovation.

Due to the slow, costly and challenging transmission of tacit knowledge (know-how), some nations cannot acquire information explaining the differences in the products they produce. Countries found a solution to deal with this issue: to specialize [12]. Adam Smith said, "the division of labor is the secret of the wealth of nations" [5]. As this concept dates back to 1776, Ricardo and his collaborators stated, "the division of labor is what allows us to access the quantity of knowledge that none of us would be able to hold individually." [6]. Products are seen as vehicles of knowledge and markets as vectors to spread that knowledge. Therefore, we can become collectively wiser. However, a person alone does not possess all the required knowledge to create a complex product. If individuals create, merge, and expand their knowledge within a country, they can develop a bundle of capabilities to create a complex economy that can sustain itself [6]. Nevertheless, not every nation possesses the same set of capabilities. Similarly to scrabble game, where players have different letters to create words, countries have individuals with different capabilities to create products. Thus, being economically complex is not that easy and depends on a country's capabilities and knowledge. Another difficulty comes from the complexity of certain types of products. Making the X-Ray machine is more complicated than producing clothing, for example. As these types of products need various capabilities' interactions, countries with high levels of complexity can usually create them. Thus, high-complex economies tend to export more diversified products, from simple to complex, whereas low-complex economies do not have that set of choices and focus on less complex products such as textiles and natural resources [50].

As economic complexity has become a hot new topic in recent decades, different ways to measure it exist. The leading two indicators are the fitness–complexity method and the method of reflections [51], which are mathematically and conceptually different [52]. Neither of the methods are perfect, and both have been criticized [53,54]. In this paper, we used the method of reflections based on the diversity and ubiquity of the products a

country export to create the economic complexity index (i.e., ECI), developed by Cesar A. Hidalgo and Ricardo Hausmann.

In previous research, we already determined that ECI improves HDI for developing countries [8]. According to Hausmann and his collaborators, in opposition to GDP per capita, economic complexity leads to higher wealth [6]. Other authors, such as Costanza, said that the ECI is a better indicator than GDP per capita in assessing economic prosperity [55]. Economic complexity is a non-income-based measure that underlines a country's hidden talents [54]. Moreover, in opposition to the human capital theory, which focuses on schooling years [56], the ECI pays attention to the productive knowledge created by economic activities.

### 2.3. Migration and Logistics: Moderators between ECI and HDI?

2.3.1. International Migration

Due to the uneven demographic transition and economic development of nations, labor markets are distorted, showing the importance of human capital. Countries' goal has always been development, and in our actual knowledge economy and information society, each country should improve collective knowledge to achieve it. However, not every country possesses the right number of skilled workers. To face this issue, the migratory attraction has been used to save time and money, but more importantly, to access the human potential formation [10]. Historically, the development of countries has always been influenced by migration [57], facilitating population redistribution. Through immigration, destination countries can meet their labor market needs in terms of age and qualification. In high-income countries, the aging population's growth and the decrease in birth have been a concern in recent decades. The "Age-selection of immigrants and their children birth within the country" is a way to balance the "age shortage of the active population" [10]. These countries can also deal with the labor market qualification needs through the training and adaptation of the migrant workers. Individuals with physical and psychological health, natural capabilities, accumulated knowledge, experience, education, and qualifications are those destination countries wish to acquire in their labor force [57,58]. This view of immigration, mainly based on Becker [57], could be seen as a commodification of human beings, which means that the access to a huge and nearly limitless number of individuals change people into products. The focus is not on commoditizing humans, which may perceive international migrants as widely available and interchangeable, but instead we see these individuals as carriers of knowledge and new working practices that have the capabilities to improve human development.

Migration flows increase when there are migrant networks [59]. Therefore, re-emigration is crucial as it expands the transfer of knowledge and know-how through joint research activities. Another advantage is that re-emigration promotes the circular migration of specialists [10] and allows the flow of knowledge, skills, technology, and capital between countries and regions of the world with different specialization, leading to innovations [60]. Entrepreneurs usually take risks and are active contributors to innovation [61], which is especially true for immigrants. Immigrants are often entrepreneurs that contribute to the economy of a country. To successfully integrate the labor market and move upward in terms of economic mobility, immigrants often see self-employment as the best choice [62]. Innovation and the improvement of technology is a determinant of human development that is related to migration. The technological structure follows a cycle, lasting 40 to 60 years, with four stages: recession, trough, expansion, and peak, related to the migration cycle [10]. In the recession stage, there is an outflow of human potential from the origin country, which is due to the individuals' desire to have better living conditions and employment. This results in a trough and partial expansion for the country of origin. At the same time, in the destination country, immigrants are adjusting and consolidating their lives. They will send remittances to their families in the home country. Second, the emigrants become accustomed to their new life. They also enjoy more employment opportunities, which is

related to the networking stage. Finally, during the stage of expansion and peak, they may decide to re-emigrate.

2.3.2. Logistics Performance

Logistics is the "part of the value chain which plans, implements, and controls the efficient flow of goods, services, and information from source to consumer" [63]. There are five basic logistics elements: logistics networking, sourcing and procurement, planning and forecasting, transportation, and distribution. Logistics is crucial as transportation, storage, and packaging issues are vital elements in the competitiveness of businesses within and between nations [64]. Through each region's synergy effects to contribute to regional logistics, a country can improve its competitive advantages and become more developed [65].

To measure logistics, the World Bank developed a new index in 2007 called the logistics performance index (LPI). The LPI provides information regarding the quality of a country's infrastructure and its logistics costs, and the customs procedures necessary for any trade [64]. It is made of six components [11] (see Appendix B). The LPI is based on a survey given to professionals working in multinational freight forwarders and express carriers [11]. Those individuals know about the decisions made in shipping routes and gateways, which is a vital element for companies regarding the location of production, suppliers' choice, and the selection of target markets. Therefore, they are qualified to provide quality and credible knowledge for the LPI data.

Effective logistics services are essential for the mobility of products and the environment. First, it provides speed, safety, and cost reduction when trading with other countries [63,64]. Furthermore, logistics is fundamental as we need access to transportations and infrastructure to produce complex products. Proper logistics developments impact the business environment, the mobility and effective use of human and other resources, and the receptivity of innovations [65]. Finally, looking for logistics systems that are performant is essential as climate change is mainly due to the increase in carbon emissions, in which logistics accounts for 13.1% [66].

## 3. Methodology

This paper has for purpose the analysis of the moderating factors, international migration (IM) and logistic performance index (LPI), between the economic complexity (ECI) and human development (HDI). The ECI was challenged due to mathematical and conceptual problems [51,52]. Hidalgo and Hausmann created the ECI+, which incorporates the difficulty to export each product, and the size of the market of that nation [67]. In this research, we use the ECI+. Governance variables—rule of law (RL), regulatory quality (RQ)—and youth unemployment (WE) were included as control variables. Social development variables—civic activism (SDCA), group inclusion (SDGI), and age dependency ratio old (SDAO)—are inserted as predictors at level 1. Gender inequality (GII) is used as a cultural predictor at level 2. Information regarding each variable is available in Appendix B.

This study focused on two groups of countries. We combined data for 117 nations from 1990 to 2017. We decided to choose the countries' level of HDI to sample the groups. The UNDP categorizes nations into four groups based on their level of human development: very high human development (0.8–1), high human development (0.7–0.799), medium human development (0.550–0.699), and low human development (0.350–0.549). As countries evolve over the years, their level of HDI changes. We calculated the average score of each country (from 1990 to 2017) and extracted their scores. A limitation of HLM methodology is that a minimum of 30 countries is required at level 2. As this requirement was not satisfied, once we separated the countries based on their HDI level, we decided to group them into two groups—high HDI (59 countries) and low HDI (55 countries). The countries' names are available in Appendix A.

In this paper, the HLM methodology was used to analyze the data. The first level represents the yearly data for each of our countries. The yearly data for the dependent, control and independent variables are inserted at that level. Like any ordinary least square (OLS) regression, it runs a standard regression analysis. Level 1 is also considered as the base level. Level 2 is the country level, where we added the country level predictor and moderators. Furthermore, to prevent collinearity issues, the group mean of the level 1 variables are added to level 2. In HLM, collinearity may appear between variables at the same level but also between cross-level correlations. Therefore, it should be dealt with carefully as it leads to biased interpretations. To solve this issue, centering the variables around their group means in level 1 and then reinserting the group mean of the variable in level 2 is a solution [68,69].

Another problem that may arise is endogeneity, which is the "correlation between the explanatory variables and the error term in a regression" [70]. Endogeneity may arise for three reasons [71]. There are two broad techniques to deal with endogeneity: ad hoc approaches and instrumental variables. This paper uses the ad hoc approach, which lags the suspected variables by one or more periods. For example, we assumed that the current values of ECI+ might be endogenous to HDI. However, the past values of ECI+ are doubtfully impacted by the same issue. Thus, in the following section, each table will present different models with different year lags. In model A, all variables are included at year t. In model B, HDI is included at year t, and all the other variables are inserted at year t − 1. In model C, HDI is included at year t, all the other variables at year t − 2. To better understand the impact of each moderator, we included them separately. Therefore, each table will provide the analysis with immigration as the moderator, and on the other side, the analysis with logistic performance. By using HLM, variables should be added to the model one at a time. Each time, the next model is compared to the previous one to determine whether the newly added items are useful to explain the variations. For more clarity, we only provide the essential tables in this paper.

## 4. Statistical Analysis

### 4.1. Null Model: Human Development as Outcome Variable

The first step in HLM is to discover whether significant differences regarding the outcome variable exist among the analysis units at the group level. Here, the prediction of the level 1 intercept (mean) of HDI as a random effect of the level 2 grouping variable is made. There are no other predictors at level 1 or level 2. The analysis of variance (ANOVA) checks the significant differences between the means and whether HDI diverges among nations. It is important to note that the null model is used as a baseline model to compare with more complex models. It is mathematically expressed in Equations (1) and (2):

$$\text{Level-1: } Y_{ij} = \beta_{0j} + \varepsilon_{ij} \tag{1}$$

$$\text{Level-2: } \beta_{0j} = \gamma_{00} + u_{0j} \tag{2}$$

$Y_{ij}$ represents HDI in *i* year and *j* country. $\beta_{0j}$ is the intercept or the average of j country's HDI. $\gamma_{00}$ is the average country mean of human development for the population of countries. $u_{0j}$ is the remaining unexplained random effect of $\beta_{0j}$, and finally, the error term $\varepsilon_{ij}$ is the unique effect associated with *i* year and *j* country. In this step, similarly to an OLS regression, a chi-square test is run. The null model was used to measure the intra-class correlation coefficient (i.e., ICC), which measures the reliability of measurements or ratings:

$$\text{ICC} = \tau_{00}/(\sigma^2 + \tau_{00}) \qquad\qquad \sigma^2 + \tau_{00} = \text{Total variance}$$

From the results of Table 1, we can see that $u_{0j}$ is different from zero, and the chi-square test reaches a significant level (*p*-value is < 0.001). For each model with different year lags, the between-group variance ($\tau_{00}$) is greater than the within-group variance ($\sigma^2$). The intra-class correlation reaches an average level of 10% for low HDI countries and

between 13 and 15% for high HDI countries. The results suggest that the model is robust across models, and further hierarchical analysis can be performed.

**Table 1.** Null model with human development index (HDI) as an outcome variable.

| Low HDI Countries | (Immigration) | | | (Logistics) | | |
|---|---|---|---|---|---|---|
| | Model A | Model B | Model C | Model A | Model B | Model C |
| $\sigma^2$ | 0.00099 | 0.00098 | 0.00096 | 0.00101 | 0.001 | 0.00098 |
| $\tau_{00}$ | 0.00902 | 0.00883 | 0.00875 | 0.00938 | 0.00919 | 0.00912 |
| ICC | 10% | 10% | 10% | 10% | 10% | 10% |
| $\gamma_{00\ HDI}$ | 0.596930 *** | 0.603349 *** | 0.608972 *** | 0.594392 *** | 0.600972 *** | 0.606800 *** |
| $u_0$ | 4011.9555 *** | 3964.8976 *** | 3990.3999 *** | 3896.8599 *** | 3854.3715 *** | 3884.7559 *** |
| Deviance | −1790.8621 | −1795.5891 | −1807.1789 | −1704.307 | −1708.8963 | −1720.2557 |
| Parameters | 2 | 2 | 2 | 2 | 2 | 2 |
| N1 | 485 | 485 | 485 | 464 | 464 | 464 |
| N2 | 41 | 41 | 41 | 39 | 39 | 39 |
| N1/N2 | 11.83 | 11.83 | 11.83 | 11.90 | 11.90 | 11.90 |
| **High HDI Countries** | **(Immigration)** | | | **(Logistics)** | | |
| | Model A | Model B | Model C | Model A | Model B | Model C |
| $\sigma^2$ | 0.00067 | 0.00061 | 0.00055 | 0.00064 | 0.00058 | 0.00053 |
| $\tau_{00}$ | 0.00384 | 0.00374 | 0000365 | 0.00362 | 0.00355 | 0.00348 |
| ICC | 15% | 14% | 13% | 15% | 14% | 13% |
| $\gamma_{00\ HDI}$ | 0.825885 *** | 0.830563 *** | 0.834841 *** | 0.830993 *** | 0.835456 *** | 0.839586 *** |
| $u_0$ | 4356.7175 *** | 4665.8716 *** | 5009.0947 *** | 3975.8619 *** | 4291.6523 *** | 4643.7044 *** |
| Deviance | −3275.3555 | −3346.9846 | −3418.6703 | −3137.7521 | −3207.2608 | −3278.2654 |
| Parameters | 2 | 2 | 2 | 2 | 2 | 2 |
| N1 | 784 | 784 | 784 | 744 | 744 | 744 |
| N2 | 52 | 52 | 52 | 49 | 49 | 49 |

$p$-value: *** < 0.001.

### 4.2. One-Way ANCOVA with Fixed Effects and Random Coefficient Regression

In the following tables, we include the control variables and predictors at level 1 while controlling for the level 2 context. It is mathematically expressed in Equations (3)–(10). In the models, $\beta_{1j}$ to $\beta_{3j}$ represent the slope of each control variable, while $\gamma_{10}$ to $\gamma_{30}$ are the fixed slopes of $\beta_{1j}$ to $\beta_{3j}$, $\gamma_{01}$ is the slope of the mean of WE. $\beta_{4j}$ to $\beta_{6j}$ represent the slope of each predictor, while $\gamma_{40}$ to $\gamma_{60}$ are the fixed slopes, $\gamma_{04}$ to $\gamma_{06}$ are the slope of the mean of SDCA, SDGI, and SDAO:

$$\text{Level-1: HDI}_{ij} = \beta_{0j} + \beta_{1j}\text{x}(\text{CV\_WE}_{ij\_centering}) + \beta_{2j}\text{x}(\text{CV\_RQ}_{ij}) + \beta_{3j}\text{x}(\text{CV\_RL}_{ij}) + \beta_{4j}\text{x}(\text{SDCA}_{ij\_centering}) + \beta_{5j}\text{x}(\text{SDGI}_{ij\_centering}) + \beta_{6j}\text{x}(\text{SDAO}_{ij\_centering}) + r_{ij} \tag{3}$$

$$\text{Level-2: } \beta_{0j} = \gamma_{00} + \gamma_{01}\text{x}(\text{CV\_WE}_{j\_mean}) + \gamma_{04}\text{x}(\text{SDCA}_{j\_mean}) + \gamma_{05}\text{x}(\text{SDGI}_{j\_mean}) + \gamma_{06}\text{x}(\text{SDAO}_{j\_mean}) + u_{0j} \tag{4}$$

$$\beta_{1j} = \gamma_{10} \tag{5}$$

$$\beta_{2j} = \gamma_{20} \tag{6}$$

$$\beta_{3j} = \gamma_{30} \tag{7}$$

$$\beta_{4j} = \gamma_{40} + u_{4j} \tag{8}$$

$$\beta_{5j} = \gamma_{50} + u_{5j} \tag{9}$$

$$\beta_{6j} = \gamma_{60} + u_{6j} \tag{10}$$

When adding each control variable and predictor, we checked that within-group variances, between-group variances, and deviances decrease compare to the previous models. From the results of Table 2, we can see that the results are robust among different models with different year low HDI countries. The explanatory power ranges from 83 to 85% at

level 1 and 83–84% at level 2 for high HDI countries. We can see that youth unemployment and group inclusion have negative short-term effects on human development in low HDI countries: the fixed slope of WE, $\gamma_{10}$, is negatively significant ($p$-value < 0.05); the fixed slope of SDGI, $\gamma_{50}$, is negatively significant ($p$-value < 0.001). Governance indicators, RQ and RL, become insignificant once the other variables are included in the model. In the long term, youth unemployment, civic activism, group inclusion, and age dependency (older people) are positively significant: the slope of the means WE ($\gamma_{01}$), and SDAO ($\gamma_{06}$) are strongly significant with $p$-values < 0.001, the slope of the mean SDCA ($\gamma_{04}$) is moderately significant ($p$-value < 0.01), and the slope of the mean SDGI ($\gamma_{05}$) is slightly significant ($p$-value < 0.05). The results are different for high HDI countries. We can see that civic activism has negative short-term effects on human development in high HDI countries: $\gamma_{40,}$ which is strongly and negatively significant ($p$-value < 0.001). Meanwhile, group inclusion, age dependency (old), and the rule of law have positive short-term effects on human development in these nations: $\gamma_{50\ SDGI}$ and $\gamma_{60\ SDAO}$ $p$-values < 0.001 $\gamma_{30\ CV\_RL}$ $p$-value < 0.05. In the long term, only civic activism ($\gamma_{04}$) has strong positive effects on HDI ($p$-value < 0.001). None of the other control variables or predictors affect the human development level of high HDI countries.

**Table 2.** One-way ANCOVA and regressions of predictors at level 1.

| Low HDI Countries | (Immigration) | | | (Logistics) | | |
|---|---|---|---|---|---|---|
| | Model A | Model B | Model C | Model A | Model B | Model C |
| $\sigma^2$ | 0.00006 | 0.00006 | 0.00007 | 0.00006 | 0.00007 | 0.00007 |
| $\tau_{00}$ | 0.00284 | 0.00288 | 0.00297 | 0.00271 | 0.00274 | 0.00285 |
| $R_1{}^2$ | 71% | 70% | 69% | 73% | 72% | 71% |
| $R_2{}^2$ | 69% | 68% | 66% | 71% | 70% | 69% |
| $\gamma_{00\ HDI}$ | −0.502963 *** | −0.469554 *** | −0.458219 *** | −0.463077 ** | −0.487142 *** | −0.493975 *** |
| $\gamma_{01\ CV\_WE}$ | 0.040769 *** | 0.039985 *** | 0.038358 *** | 0.036858 *** | 0.036063 *** | 0.035597 *** |
| $\gamma_{04\ SDCA}$ | 0.857604 ** | 0.843859 ** | 0.835479 ** | 0.782236 ** | 0.869399 *** | 0.886099 *** |
| $\gamma_{05\ SDGI}$ | 0.385315 * | 0.360866 * | 0.353912 * | 0.395966 * | 0.395075 *** | 0.395275 *** |
| $\gamma_{06\ SDAO}$ | 0.211235 *** | 0.207757 *** | 0.210826 *** | 0.213927 *** | 0.209341 *** | 0.211952 *** |
| $\gamma_{10\ CV\_WE}$ | −0.005005 * | −0.004158 + | −0.003690 + | −0.006462 ** | −0.005317 * | −0.004779 * |
| $\gamma_{20\ CV\_RQ}$ | 0.542 | 0.482 | 0.186 | 0.875 | 0.810 | 0.437 |
| $\gamma_{30\ CV\_RL}$ | 0.557 | 0.537 | 0.355 | 0.013193 * | 0.108 | 0.146 |
| $\gamma_{40\ SDCA}$ | 0.237 | 0.190 | 0.148 | 0.526 | 0.366 | 0.224 |
| $\gamma_{50\ SDGI}$ | −0.297087 *** | −0.327135 *** | −0.323316 ** | −0.275625 *** | −0.326874 *** | −0.329615 *** |
| $\gamma_{60\ SDAO}$ | 0.900 | 0.959 | 0.718 | 0.921 | 0.898 | 0.772 |
| $u_0$ | 15997.7858 *** | 15792.1472 *** | 15916.4121 *** | 15922.3070 *** | 15197.7436 *** | 14589.2910 *** |
| $u_4$ | 1394.3432 *** | 1178.3105 *** | 912.6450 *** | 1217.7934 *** | 1070.2530 *** | 847.7841 *** |
| $u_5$ | 140.1426 *** | 147.0484 *** | 163.5485 *** | 73.7099 *** | 77.7731 *** | 82.5788 *** |
| $u_6$ | 150.0766 *** | 162.5342 *** | 156.9446 *** | 124.2618 *** | 143.8962 *** | 137.1385 *** |
| Deviance | −2728.759 | −2702.9751 | −2681.3747 | −2637.6813 | −2600.6854 | −2569.9518 |
| Parameters | 12 | 12 | 12 | 12 | 12 | 12 |
| **High HDI Countries** | **(Immigration)** | | | **(Logistics)** | | |
| | Model A | Model B | Model C | Model A | Model B | Model C |
| $\sigma^2$ | 0.00007 | 0.00006 | 0.00006 | 0.00006 | 0.00006 | 0.00006 |
| $\tau_{00}$ | 0.0006 | 0.00061 | 0.00062 | 0.00059 | 0.00061 | 0.00061 |
| $R_1{}^2$ | 85% | 85% | 84% | 85% | 84% | 83% |
| $R_2{}^2$ | 84% | 84% | 83% | 84% | 83% | 83% |
| $\gamma_{00\ HDI}$ | 0.516794 *** | 0.517156 *** | 0.522240 *** | 0.518216 *** | 0.516922 *** | 0.520149 *** |
| $\gamma_{01\ CV\_WE}$ | 0.951 | 0.980 | 0.981 | 0.874 | 0.952 | 0.878 |
| $\gamma_{04\ SDCA}$ | 0.478652 *** | 0.487425 *** | 0.482304 *** | 0.447109 *** | 0.459734 *** | 0.461209 *** |
| $\gamma_{05\ SDGI}$ | 0.752 | 0.891 | 0.972 | 0.605 | 0.694 | 0.795 |
| $\gamma_{06\ SDAO}$ | 0.883 | 0.650 | 0.436 | 0.892 | 0.566 | 0.406 |
| $\gamma_{10\ CV\_WE}$ | −0.005589 + | 0.301 | 0.708 | 0.128 | 0.410 | 0.690 |
| $\gamma_{20\ CV\_RQ}$ | 0.430 | 0.498 | 0.581 | 0.243 | 0.387 | 0.415 |
| $\gamma_{30\ CV\_RL}$ | 0.018540 * | 0.015875 * | 0.013952 + | 0.019711 * | 0.016453 + | 0.108 |

**Table 2.** *Cont.*

| High HDI Countries | (Immigration) | | | (Logistics) | | |
|---|---|---|---|---|---|---|
| | Model A | Model B | Model C | Model A | Model B | Model C |
| $\gamma_{40}$ SDCA | −0.287405 *** | −0.290084 *** | −0.292491 *** | −0.185891 *** | −0.168679 *** | −0.150195 *** |
| $\gamma_{50}$ SDGI | 0.095755 ** | 0.085368 ** | 0.075793 * | 0.134866 *** | 0.122484 *** | 0.110676 *** |
| $\gamma_{60}$ SDAO | 0.235920 *** | 0.220994 *** | 0.205534 *** | 0.232605 *** | 0.223418 *** | 0.212095 *** |
| $u_0$ | 6168.7710 *** | 6696.2822 *** | 6982.4773 *** | 5978.8505 *** | 6358.0342 *** | 6707.9002 *** |
| $u_4$ | 522.6428 *** | 474.6855 *** | 423.2662 *** | 508.8023 *** | 468.4861 *** | 434.2481 *** |
| $u_5$ | 281.9977 *** | 308.1606 *** | 349.4850 *** | 160.9590 *** | 143.7397 *** | 146.8742 *** |
| $u_6$ | 239.8716 *** | 222.9460 *** | 211.6456 *** | 260.0511 *** | 234.7836 *** | 225.1805 *** |
| Deviance | −4663.1381 | −4710.6553 | −4731.7529 | −4539.509 | −4579.1434 | −4616.6333 |
| Parameters | 12 | 12 | 12 | 12 | 12 | 12 |

*p*-value: + < 0.1, * < 0.5, ** < 0.01, *** < 0.001.

### 4.3. Random Coefficient Regression: The Effects of Economic Complexity

We included ECI+ as a predictor of HDI. In the following equation, $\beta_{7j}$ is the slope of ECI+, $\gamma_{70}$ is the fixed slope of $\beta_{7j}$, and $\gamma_{07}$ is the slope of the mean of ECI+. The model is mathematically expressed as follows in Equations (11)–(19):

$$\text{Level-1: HDI}_{ij} = \beta_{0j} + \beta_{1j}x(\text{CV\_WE}_{ij\_centering}) + \beta_{2j}x(\text{CV\_RQ}_{ij}) + \beta_{3j}x(\text{CV\_RL}_{ij}) + \beta_{4j}x(\text{SDCA}_{ij\_centering}) + \beta_{5j}x(\text{SDGI}_{ij\_centering}) + \beta_{6j}x(\text{SDAO}_{ij\_centering}) + \beta_{7j}x(\text{ECI+}_{ij\_centering}) + r_{ij} \tag{11}$$

$$\text{Level-2: } \beta_{0j} = \gamma_{00} + \gamma_{01}x(\text{CV\_WE}_{j\_mean}) + \gamma_{04}x(\text{SDCA}_{j\_mean}) + \gamma_{05}x(\text{SDGI}_{j\_mean}) + \gamma_{06}x(\text{SDAO}_{j\_mean}) + \gamma_{07}x(\text{ECI+}_{j\_mean}) + u_{0j} \tag{12}$$

$$\beta_{1j} = \gamma_{10} \tag{13}$$

$$\beta_{2j} = \gamma_{20} \tag{14}$$

$$\beta_{3j} = \gamma_{30} \tag{15}$$

$$\beta_{4j} = \gamma_{40} + u_{4j} \tag{16}$$

$$\beta_{5j} = \gamma_{50} + u_{5j} \tag{17}$$

$$\beta_{6j} = \gamma_{60} + u_{6j} \tag{18}$$

$$\beta_{7j} = \gamma_{70} + u_{7j} \tag{19}$$

Table 3 shows that ECI+ is insignificant in the short term but has a weak positive effect on HDI in the long term for low HDI countries: the *p*-value of the slope's average, $\gamma_{07}$, is significant (*p*-value < 0.1). In other words, economic complexity only impacts the human development level of low countries in the long term. For high HDI countries the results are different. ECI+ has negative effects in the short term: the fixed slope, $\gamma_{70}$, is weakly significant (*p*-value < 0.1). Under the immigration model, the mean slope of $\gamma_{07}$ is insignificant, whereas it is weakly and positively significant (*p*-value < 0.1) under the logistics model. The results induce that economic complexity first negatively impacts the human development level in high HDI countries, which will become positive in the long term under certain conditions. However, we cannot conclude yet, as gender inequality should be included as a country-level predictor and may impact the long-term effects of HDI. Similarly, the moderators may also influence the impact of ECI+ on HDI.

**Table 3.** Regression of ECI+.

| Low HDI Countries | (Immigration) | | | (Logistics) | | |
|---|---|---|---|---|---|---|
| | **Model A** | **Model B** | **Model C** | **Model A** | **Model B** | **Model C** |
| $\sigma^2$ | 0.00005 | 0.00005 | 0.00006 | 0.00005 | 0.00005 | 0.00006 |
| $\tau_{00}$ | 0.00283 | 0.00273 | 0.00278 | 0.00263 | 0.00266 | 0.00274 |
| $R_1^2$ | 71% | 72% | 71% | 74% | 73% | 72% |
| $R_2^2$ | 69% | 69% | 68% | 72% | 71% | 70% |
| $\gamma_{00}$ HDI | −0.276712 | 0.135 | 0.163 | −0.323770+ | 0.112 | 0.175 |
| $\gamma_{01}$ CV_WE | 0.041545 *** | 0.037970 *** | 0.037437 *** | 0.032500 ** | 0.031903 ** | 0.033506 ** |
| $\gamma_{04}$ SDCA | 0.624803 * | 0.647981 * | 0.631928 * | 0.713923 ** | 0.698356 * | 0.637526 * |
| $\gamma_{05}$ SDGI | 0.268 | 0.300915 + | 0.288903 + | 0.143 | 0.131 | 0.127 |
| $\gamma_{06}$ SDAO | 0.204139 *** | 0.174265 *** | 0.175777 *** | 0.198355 *** | 0.188005 *** | 0.185050 *** |
| $\gamma_{07}$ ECI+ | 0.028234 + | 0.027362 + | 0.028036 + | 0.026053 + | 0.025561 + | 0.026949 + |
| $\gamma_{10}$ CV_WE | −0.006371 ** | −0.005016 * | −0.003624 * | −0.006825 ** | −0.005866 ** | −0.004507 * |
| $\gamma_{20}$ CV_RQ | 0.482 | 0.623 | 0.325 | 0.876 | 0.912 | 0.544 |
| $\gamma_{30}$ CV_RL | 0.252 | 0.340 | 0.158 | 0.014498 ** | 0.012897 * | 0.014007 * |
| $\gamma_{40}$ SDCA | 0.687 | 0.146 | −0.406152 * | 0.303 | 0.237 | −0.276279 + |
| $\gamma_{50}$ SDGI | −0.136179 * | −0.246886 *** | −0.276110 ** | −0.251371 *** | −0.281011 *** | −0.292274 *** |
| $\gamma_{60}$ SDAO | 0.894 | 0.602 | 0.669 | 0.669 | 0.721 | 0.716 |
| $\gamma_{70}$ ECI+ | 0.254 | 0.867 | 0.727 | 0.427 | 0.700 | 0.654 |
| $u_0$ | 21518.3112 *** | 19052.7810 *** | 17983.4107 *** | 21390.6781 *** | 19371.5446 *** | 17504.0685 *** |
| $u_4$ | 808.9027 *** | 722.2961 *** | 567.6924 *** | 739.0508 *** | 649.7427 *** | 540.4783 *** |
| $u_5$ | 78.9834 *** | 78.5213 *** | 94.6734 *** | 51.5273 *** | 46.0329 *** | 55.1069 *** |
| $u_6$ | 91.0402 *** | 117.1158 *** | 120.8872 *** | 113.2219 *** | 125.7335 *** | 119.8428 *** |
| $u_7$ | 91.94806 | 71.61 *** | 56.3745 *** | 86.3465 *** | 67.9650 *** | 50.7909 *** |
| Deviance | −2775.3368 | −2746.1737 | −2721.6158 | −2679.0208 | −2644.1737 | −2611.7338 |
| Parameters | 14 | 14 | 14 | 14 | 14 | 14 |
| **High HDI Countries** | **(Immigration)** | | | **(Logistics)** | | |
| | **Model A** | **Model B** | **Model C** | **Model A** | **Model B** | **Model C** |
| $\sigma^2$ | 0.00005 | 0.00004 | 0.00004 | 0.00004 | 0.00004 | 0.00004 |
| $\tau_{00}$ | 0.00058 | 0.00059 | 0.00059 | 0.00057 | 0.00058 | 0.00058 |
| $R_1^2$ | 86% | 86% | 85% | 86% | 85% | 85% |
| $R_2^2$ | 85% | 84% | 84% | 84% | 84% | 83% |
| $\gamma_{00}$ HDI | 0.540434 *** | 0.536856 *** | 0.540884 *** | 0.549288 *** | 0.542606 *** | 0.538077 *** |
| $\gamma_{01}$ CV_WE | 0.643 | 0.570 | 0.486 | 0.555 | 0.528 | 0.333 |
| $\gamma_{04}$ SDCA | 0.430624 *** | 0.442072 *** | 0.433072 *** | 0.408389 *** | 0.421770 *** | 0.417314 *** |
| $\gamma_{05}$ SDGI | 0.794 | 0.777 | 0.770 | 0.838 | 0.840 | 0.743 |
| $\gamma_{06}$ SDAO | 0.636 | 0.653 | 0.760 | 0.594 | 0.779 | 0.779 |
| $\gamma_{07}$ ECI+ | 0.111 | 0.130 | 0.120 | 0.015637 + | 0.016594 + | 0.018402 + |
| $\gamma_{10}$ CV_WE | −0.007462 ** | 0.104 | 0.948 | −0.006422 * | 0.164 | 0.876 |
| $\gamma_{20}$ CV_RQ | 0.507 | 0.453 | 0.521 | 0.147 | 0.230 | 0.187 |
| $\gamma_{30}$ CV_RL | 0.019002 *** | 0.015608 ** | 0.013984 * | 0.019354 *** | 0.015372 * | 0.013004 + |
| $\gamma_{40}$ SDCA | −0.246335 ** | −0.232098 ** | −0.237297 * | −0.132932 *** | −0.126963 *** | −0.116912 *** |
| $\gamma_{50}$ SDGI | 0.074414 * | 0.066511 * | 0.059993 + | 0.091152 *** | 0.089119 *** | 0.085014 *** |
| $\gamma_{60}$ SDAO | 0.190761 *** | 0.189561 *** | 0.179313 *** | 0.189559 *** | 0.187661 *** | 0.183442 *** |
| $\gamma_{70}$ ECI+ | −0.029527 + | −0.032414 * | −0.031131 + | −0.031158 + | −0.034151 + | 0.103 |
| $u_0$ | 8189.7020 *** | 9079.0627 *** | 9217.6888 *** | 8500.1171 *** | 9069.1074 *** | 9490.2241 *** |
| $u_4$ | 327.9539 *** | 332.0318 *** | 307.0721 *** | 338.6365 *** | 319.5659 *** | 311.0637 *** |
| $u_5$ | 258.4150 *** | 262.8035 *** | 292.6618 *** | 156.1952 *** | 153.7890 *** | 168.7762 *** |
| $u_6$ | 210.8777 *** | 197.3610 *** | 190.7771 *** | 239.9130 *** | 216.0536 *** | 212.1180 *** |
| $u_7$ | 210.8699 *** | 214.4034 *** | 198.2651 *** | 231.7406 *** | 222.3070 *** | 209.4435 *** |
| Deviance | −4771.9904 | −4812.0524 | −4819.3063 | −4677.734 | −4707.7203 | −4734.4413 |
| Parameters | 14 | 14 | 14 | 14 | 14 | 14 |

*p*-value: + < 0.1, * < 0.5, ** < 0.01, *** < 0.001.

### 4.4. Country-Level Predictor and Moderators

GII is included as a predictor in level 2, and LPI and IM are added as moderators. The equation is mathematically expressed in Equations (20)–(28), where $\gamma_{08}$ represents the slope of the predictor GII, $\gamma_{71}$ the moderator IM or LPI:

$$\text{Level-1: HDI}_{ij} = \beta_{0j} + \beta_{1j}x(\text{CV\_WE}_{ij\_centering}) + \beta_{2j}x(\text{CV\_RQ}_{ij}) + \beta_{3j}x(\text{CV\_RL}_{ij}) + \beta_{4j}x(\text{SDCA}_{ij\_centering}) + \\ \beta_{5j}x(\text{SDGI}_{ij\_centering}) + \beta_{6j}x(\text{SDAO}_{ij\_centering}) + \beta_{7j}x(\text{ECI+}_{ij\_centering}) + r_{ij} \tag{20}$$

$$\text{Level-2: } \beta_{0j} = \gamma_{00} + \gamma_{01}x(\text{CV\_WE}_{j\_mean}) + \gamma_{04}x(\text{SDCA}_{j\_mean}) + \gamma_{05}x(\text{SDGI}_{j\_mean}) + \gamma_{06}x(\text{SDAO}_{j\_mean}) + \\ \gamma_{07}x(\text{ECI+}_{j\_mean}) + \gamma_{08}x(\text{GII+}_{j\_mean}) + u_{0j} \tag{21}$$

$$\beta_{1j} = \gamma_{10} \tag{22}$$

$$\beta_{2j} = \gamma_{20} \tag{23}$$

$$\beta_{3j} = \gamma_{30} \tag{24}$$

$$\beta_{4j} = \gamma_{40} + u_{4j} \tag{25}$$

$$\beta_{5j} = \gamma_{50} + u_{5j} \tag{26}$$

$$\beta_{6j} = \gamma_{60} + u_{6j} \tag{27}$$

$$\beta_{7j} = \gamma_{70} + \gamma_{71}x(\text{IM}_{\_mean} \text{ or LPI}_{\_mean}) + u_{7j} \tag{28}$$

From both Table 4, we can see that gender inequality negatively impacts human development. The country-level effect of GII has a moderate impact on HDI for both low and high HDI countries: the slope of GII, $\gamma_{08}$, has a $p$-value < 0.01. Once gender inequality is included in the model, the previous long-term slight effects of ECI totally disappear for high HDI nations.

**Table 4.** Full model.

| Low HDI Countries | (Immigration) | | | (Logistics) | | |
|---|---|---|---|---|---|---|
| | Model A | Model B | Model C | Model A | Model B | Model C |
| $\sigma^2$ | 0.00005 | 0.00006 | 0.00006 | 0.00005 | 0.00006 | 0.00006 |
| $\tau_{00}$ | 0.0024 | 0.00236 | 0.00235 | 0.00237 | 0.00243 | 0.0024 |
| $R_1^2$ | 76% | 75% | 75% | 77% | 76% | 76% |
| $R_2^2$ | 74% | 73% | 73% | 75% | 74% | 74% |
| $\gamma_{00}$ HDI | 0.623 | 0.193 | 0.477 | 0.360 | 0.340343 + | 0.124 |
| $\gamma_{01}$ CV_WE | 0.035763 *** | 0.034756 ** | 0.030447 ** | 0.024110 * | 0.024478 * | 0.022751 * |
| $\gamma_{04}$ SDCA | 0.643036 ** | 0.584065 ** | 0.717235 ** | 0.581131 ** | 0.557458 * | 0.681944 ** |
| $\gamma_{05}$ SDGI | 0.959 | 0.874 | 0.589 | 0.566 | 0.934 | 0.877 |
| $\gamma_{06}$ SDAO | 0.139333 *** | 0.106934 *** | 0.107014 ** | 0.127876 *** | 0.084559 *** | 0.085690 * |
| $\gamma_{07}$ ECI+ | 0.228 | 0.288 | 0.249 | 0.587 | 0.501 | 0.400 |
| $\gamma_{08}$ GII | −0.303010 ** | −0.361219 ** | −0.348120 ** | −0.360431 ** | −0.428569 *** | −0.431185 *** |
| $\gamma_{10}$ CV_WE | −0.005857 * | −0.004976 * | −0.003309 + | −0.006319 ** | −0.005554 ** | −0.004316 * |
| $\gamma_{20}$ CV_RQ | 0.441 | 0.492 | 0.327 | 0.766 | 0.826 | 0.569 |
| $\gamma_{30}$ CV_RL | 0.282 | 0.362 | 0.205 | 0.014458 ** | 0.013072 * | 0.012740 * |
| $\gamma_{40}$ SDCA | 0.545 | 0.324 | −0.412807 * | 0.649 | 0.415 | −0.277848 + |
| $\gamma_{50}$ SDGI | −0.151988 ** | −0.191028 ** | −0.280349 *** | −0.187682 *** | −0.233597 *** | −0.302112 *** |
| $\gamma_{60}$ SDAO | 0.915 | 0.873 | 0.755 | 0.817 | 0.821 | 0.796 |
| $\gamma_{70}$ ECI+ | 0.175 | 0.451 | 0.872 | 0.167377 ** | 0.144268 ** | 0.506 |
| $\gamma_{71}$ Moderator | 0.227 | 0.541 | 0.526 | −0.068276 ** | −0.058589 * | 0.551 |
| $u_0$ | 15845.0010 *** | 14055.5373 *** | 13584.0653 *** | 15692.0723 *** | 14701.9404 *** | 13483.6594 *** |
| $u_4$ | 817.7636 *** | 693.2346 *** | 577.2928 *** | 778.9323 *** | 668.5099 *** | 562.6835 *** |
| $u_5$ | 77.6565 *** | 71.3696 *** | 95.3447 *** | 52.3654 *** | 43.8915 *** | 54.9621 *** |
| $u_6$ | 92.4756 *** | 95.9295 *** | 123.0420 *** | 92.3633 *** | 102.4796 *** | 123.8075 *** |
| $u_7$ | 82.7678 *** | 67.0436 *** | 53.5872 *** | 71.9771 *** | 64.1553 *** | 50.4711 *** |
| Deviance | −2770,535 | −2742,5516 | −2716,8832 | −2681,7835 | −2647,8674 | −2611,0861 |
| Parameters | 16 | 16 | 16 | 16 | 16 | 16 |

**Table 4.** *Cont.*

| High HDI Countries | (Immigration) | | | (Logistics) | | |
|---|---|---|---|---|---|---|
| | Model A | Model B | Model C | Model A | Model B | Model C |
| $\sigma^2$ | 0.00005 | 0.00004 | 0.00004 | 0.00004 | 0.00004 | 0.00004 |
| $\tau_{00}$ | 0.00046 | 0.00046 | 0.00046 | 0.00046 | 0.00045 | 0.00045 |
| $R_1{}^2$ | 89% | 89% | 88% | 88% | 88% | 88% |
| $R_2{}^2$ | 88% | 88% | 87% | 87% | 87% | 87% |
| $\gamma_{00}$ HDI | 0.693545 *** | 0.703952 *** | 0.711196 *** | 0.717673 *** | 0.729827 *** | 0.731924 *** |
| $\gamma_{01}$ CV_WE | 0.907 | 0.828 | 0.848 | 0.885 | 0.796 | 0.937 |
| $\gamma_{04}$ SDCA | 0.367406 *** | 0.369041 *** | 0.356818 *** | 0.315151 *** | 0.315851 *** | 0.307056 *** |
| $\gamma_{05}$ SDGI | 0.971 | 0.897 | 0.889 | 0.967 | 0.997 | 0.960 |
| $\gamma_{06}$ SDAO | −0.019260 ** | −0.018635 ** | −0.017182 * | −0.018500 ** | −0.017527 ** | −0.017643 ** |
| $\gamma_{07}$ ECI+ | 0.445 | 0.529 | 0.541 | 0.421 | 0.429 | 0.327 |
| $\gamma_{08}$ GII | −0.164044 ** | −0.172050 ** | −0.172897 ** | −0.174603 ** | −0.191595 *** | −0.197220 *** |
| $\gamma_{10}$ CV_WE | −0.007633 ** | −0.004852 + | 0.853 | −0.006998 * | 0.118 | 0.949 |
| $\gamma_{20}$ CV_RQ | 0.654 | 0.610 | 0.674 | 0.204 | 0.306 | 0.251 |
| $\gamma_{30}$ CV_RL | 0.016010 ** | 0.012884 * | 0.011290 + | 0.016433 ** | 0.012643 * | 0.113 |
| $\gamma_{40}$ SDCA | −0.243731 ** | −0.226369 * | −0.228385 * | −0.119463 ** | −0.110382 ** | −0.100019 ** |
| $\gamma_{50}$ SDGI | 0.076149 ** | 0.066455 * | 0.059046 + | 0.084401 ** | 0.083610 ** | 0.080346 *** |
| $\gamma_{60}$ SDAO | 0.190245 *** | 0.188317 *** | 0.178569 *** | 0.189377 *** | 0.188180 *** | 0.184754 *** |
| $\gamma_{70}$ ECI+ | 0.045778 * | 0.043501 * | 0.037342 + | 0.266697 ** | 0.253721 * | 0.233801 * |
| $\gamma_{71}$ Moderator | −0.044289 *** | −0.044574 *** | −0.040320 ** | −0.089319 ** | −0.086110 ** | −0.078964 * |
| $u_0$ | 5396.2038 *** | 5773.4580 *** | 5802.4855 *** | 5785.9657 *** | 5880.3432 *** | 6034.6530 *** |
| $u_4$ | 331.2791 *** | 333.7251 *** | 309.9527 *** | 343.6711 *** | 325.0528 *** | 317.1481 *** |
| $u_5$ | 254.9266 *** | 255.2222 *** | 279.3174 *** | 157.6940 *** | 155.9764 *** | 171.5476 *** |
| $u_6$ | 209.6875 *** | 195.7076 *** | 188.7161 *** | 234.4126 *** | 210.9014 *** | 207.0975 *** |
| $u_7$ | 212.0587 *** | 213.3933 *** | 199.5146 *** | 210.1653 *** | 208.4370 *** | 203.5131 *** |
| Deviance | −4713.1259 | −4824.0795 | −4829.6843 | −4688.9153 | −4720.6661 | −4747.3521 |
| Parameters | 16 | 16 | 16 | 16 | 16 | 16 |

*p*-value: + < 0.1, * < 0.5, ** < 0.01, *** < 0.001.

Under the models that include immigration as a moderator, the effects of ECI+ on HDI level in low countries disappear for both the short and long term. For this group of countries, immigration is not a significant moderator. The story is different for high HDI countries. The fixed slope of ECI+, $\gamma_{70}$, is slightly significant (*p*-value < 0.05), but more importantly, its impact on HDI changed with immigration. IM, $\gamma_{71}$, has a strong negative impact (*p*-value < 0.001) on the relationship between ECI+ and HDI. Thus, the previously negative effects of ECI+ become positive once immigration is included as a moderator. Similar behavior can be seen when logistics performance is used as a moderator. From Table 4, the slope of the mean LPI, $\gamma_{71}$, is negative and moderately (*p*-value < 0.01) affects the relationship between ECI+ and HDI under both groups of countries. However, with different year lags, the effects decrease (*p*-value > 0.01). More importantly, the effects of ECI+ on HDI become moderately and positively significant.

The moderating effects are analyzed through: $\beta_{7j} = \gamma_{70} + \gamma_{71} {}^*(LPI_j \text{ or } IM_j) + u_{7j}$. From the results of Table 5, we can see that the value of LPI, under low HDI countries, becomes negative towards the maximum value. Similarly, IM and LPI's value, under high HDI countries, become negative towards the maximum value. In other words, in the beginning, IM and LPI strengthen the relationship between ECI+ on HDI. However, too many migrants and a too high performance of logistics worsen the strength between ECI+ and HDI at a certain point.

**Table 5.** Analyzing the slope of the moderator logistics performance index (LPI) and international migration (IM).

| Low HDI Countries | Model A | Model B | Model C |
| --- | --- | --- | --- |
| $\gamma_{70}$ ECI+ | 0.167377 | 0.144268 | 0.036182 |
| $\gamma_{71}$ LPI | −0.068276 | −0.058589 | −0.013779 |
| If LPI min = 2.17 | 0.01921808 | 0.01712987 | 0.00628157 |
| If LPI max = 3.59 | −0.07773384 | −0.06606651 | −0.01328461 |
| **High HDI Countries** | **Model A** | **Model B** | **Model C** |
| $\gamma_{70}$ ECI+ | 0.045778 | 0.043501 | 0.037342 |
| $\gamma_{71}$ IM | −0.044289 | −0.044574 | −0.04032 |
| If IM min = −1.56 | 0.11486884 | 0.11303644 | 0.1002412 |
| If IM max = 4.08 | −0.13492112 | −0.13836092 | −0.1271636 |
| $\gamma_{70}$ ECI+ | 0.266697 | 0.253721 | 0.233801 |
| $\gamma_{71}$ LPI | −0.089319 | −0.08611 | −0.078964 |
| If LPI min = 2.28 | 0.06304968 | 0.0573902 | 0.05376308 |
| If LPI max = 4.16 | −0.10487004 | −0.1044966 | −0.09468924 |

## 5. Conclusions

### 5.1. Discussion

From this paper's results, we determined that human development variations in both low HDI countries and high HDI countries are explained by different control variables—youth unemployment (WE), rule of law (RL), and regulatory quality (RQ); predictors at level 1—economic complexity (ECI+), group inclusion (SDGI), age dependency old (SDAO), and civic activism (SDCA); a predictor at level 2—gender inequality (GII); and two moderators—international migration (IM) and logistics performance (LPI). The framework of the model is shown on Figure 1. For low HDI countries, in the short term, WE and SDGI negatively affect human development. RL only positively impacts HDI levels under the LPI model. In the long term, SDCA, WE, and SDAO positively affect HDI. For high HDI countries, WE and SDCA negatively influence HDI in the short term, while RL, SDGI, and SDAO positively impact HDI. In the long term, SDAO has a negative effect, and SDCA has a positive impact.

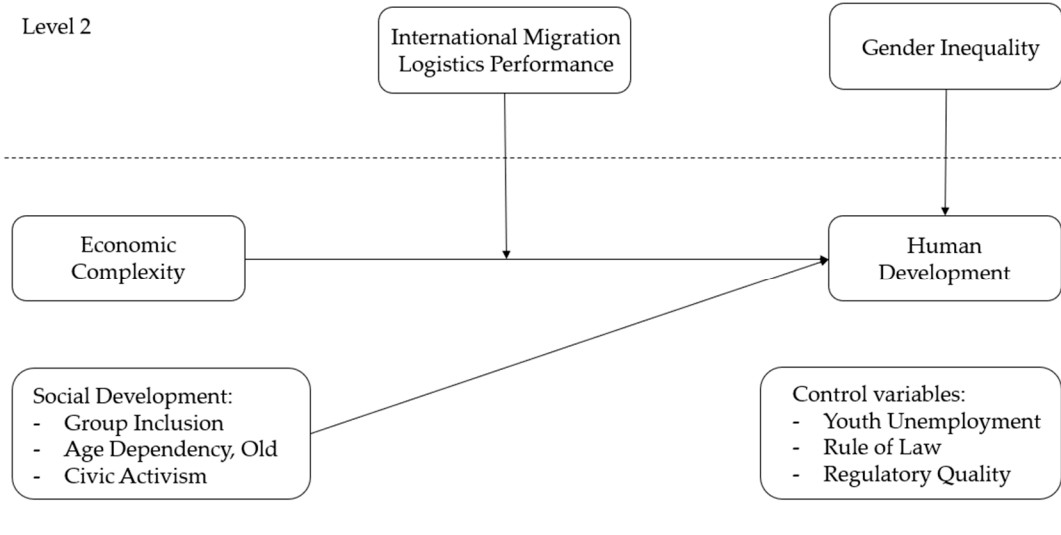

**Figure 1.** Graphical representation of the model.

In other words, in low HDI countries, youth unemployment first negatively impacts the level of human development, but in the long term, these effects become positive. For

high HDI countries, the short-term adverse effects become insignificant in the long term. Youth unemployment is usually detrimental to human development as it means that there may not be job opportunities for the young generation, so the harmful effects of youth unemployment were expected. The long-term positive effects of youth unemployment in low HDI countries can be explained by the fact that the young generation has access to extended education and postpone their entrance into the labor market. From 1993 to 2017, the gross enrolment in tertiary education rose from 14 to 38% [35]. Youths may also be engaged in volunteer work or unpaid trainee work. However, there is still some apprehension regarding the decline of youth employment, such as the perceived inability of young workers to find an appropriate job. Since the global crisis in 2009, young workers have encountered difficulties in finding decent jobs. "Being in employment does not always guarantee a decent living" [35]. In low-income and high-income countries, young individuals who have a job still experience poverty. In the EU, this pattern is explained by the fact that three-quarters of young workers mainly have informal jobs. This type of job is not illegal; however, it is not regulated, and no taxes are paid. Therefore, the workers will not be subject to national labor legislation, income taxation, or social protection [72]. Furthermore, with the advancement in new technologies, young workers should acquire high levels of technical and soft skills. It is primordial for young workers to adapt quickly to these new opportunities. Policy makers should provide proper structures and policies to help youths as most countries are encountering an aging society [31] and a need to replace older generations.

The rule of law only has a positive short-term effect for high HDI countries, and for low HDI countries under the LPI model. As the rule of law is "the extent to which agents have confidence in and abide by the rules of society" [44], protecting private property rights and making long-term contract enforcement easier will increase investments and development. Therefore, the quality of institutions strongly impacts the level of development [41]. In other words, the rule of law is an incentive structure established by formal institutions that is essential as it guarantees police enforcement and judicial sanctions, providing a safe environment for citizens and their well-being.

Even though individuals worldwide have better access to education and health, there are still social and economic inequalities. The 2030 Agenda for Sustainable Development (GA UN) has for goal to "leave no one behind" [73]. Therefore, the main goal is to create societies that promote equality, social inclusion, and social justice. However, some of our social development indicators negatively affect the level of human development. The inclusion of minorities has positive short-term effects in high HDI countries, but these effects are negative in low HDI nations. It may be explained that when a country encounters economic difficulties or waves of immigration, local citizens tend to have a negative perception of migrants and other ethnic groups, as they believe that they are taking away their jobs and are potentially involved in criminal activities [28]. To deal with this misconception, a country should show tolerance and respect through its institutions, history, and values.

Civic activism has a short-term negative impact in high HDI countries, but none in low HDI nations. However, in the long term, both groups of countries experience positive long-term effects. Civic engagement, which is participation in political life (voting), engagement in the community, and neighborhood behaviors, is usually associated with positive effects such as philanthropy, health, and well-being. However, adverse outcomes may also arise due to civic engagement, which could be due to how the community uses the resources [74]. Another explanation could be that individuals may not be given some job opportunities if the access to specific jobs is prioritized to a particular ethnic group [75].

Another social development indicator was the age dependency ratio of older people. It has short-term positive effects in high HDI countries. In the long term, the effects are positive for low HDI countries and negative for HDI nations. Aging may be seen as an encumbrance to our modern societies. However, there are both problems and opportunities related to aging. Issues related to aging will need better social care and healthcare services,

leading to more healthcare expenditure in both low- and high-income countries [76]. Furthermore, with the development of technologies, the way we work is changing. Older people should adapt to new processes such as automation, the internet, and the digital economy. To contribute to the productivity of the economy, the older workforce should undergo re-skilling and training. However, instead of seeing older people as a cost of care provision, we should see them as an achievement of the progress we have made in medical and public health. Longevity has been made possible through development in education and technology [31,76]. Moreover, economic inequalities may reduce with an aging population, even though higher mean age is usually related to higher dependency ratios [73]. The idea is that the labor supply decreases when the population becomes older. To counteract the losses in terms of human capital, companies' solutions could be to invest in technologies that make labor more productive. According to Gotmarks and his colleagues, with the decrease in fertility rates, families are now smaller, allowing a decrease in economic inequalities as a better investment is made for each child's education [32]. Moreover, older individuals are usually active community members. In low-income countries, older generations usually take care of their grandchildren, enabling parents to work more [77].

The primary purpose of this research is to comprehend the relationship between economic complexity and human development. When ECI+ is included without the country level predictor and the moderators, it is insignificant in the short term and has weak positive effects in the long term in low HDI countries. For high HDI countries, ECI+ has short negative weak effects, that are insignificant in the long term under the immigration model, and slightly positive under the logistics model. Before jumping to a conclusion, gender inequality and the moderators were included in the model as the effects of ECI+ on HDI may be influenced by international migration and logistics performance. In the long term, the impact of ECI+ disappears when gender inequality is added to the model. As we were expecting, gender inequality harms both groups of nations. If women do not have the same opportunities to be knowledgeable, healthy, with proper jobs, and get access to political positions, it is detrimental to human development. By providing better equality among genders, with reforms in women's participation in social and political life, countries guarantee better human rights and living standards for all of their citizens [24]. It is important to remember that both men and women provide for the development of their countries.

In the short term, the effects of ECI+ on HDI become slightly significant for low HDI countries, but only under the LPI model. The effects are also favorable for high HDI countries under both models of the moderator. The results suggest that immigration only plays as a moderator for high HDI countries, while logistics performance plays a moderator for both groups of countries. The impact of international migration is not as significant as logistics performance but is still primordial as both moderators make the ECI+ positively significant (Table 4). At first, the increase in international migration and the better performance of logistics positively affect the impact of ECI+ on HDI. However, at a certain point, international migration and logistics performance hinder the relationship between our two main variables (Table 5). International migration is competitive among high-income countries. They try to attract individuals with knowledge, skills, and experience that can contribute to elaborating new technologies and innovations [57,58] that can be sustainable in providing for the satisfaction of their citizens' human needs. In high-income countries, which are considered consumer economies, to satisfy individuals' needs, industries are usually resource-and-energy-intensive, leading to a large footprint. The corporate footprint may be due to mature technologies. Therefore, entrepreneurs and innovators become essential as they create radical new technologies [46]. The turning point of international migration into a negative impact on human development maybe because our indicator does not separate migrants with low or high skills. Therefore, once immigration increases a lot in this paper, we do not know if the individuals possess the right capabilities to innovate and increase human development. As international migration

directly impacts the origin country's labor market structure [78], it alters the industrial structure and impacts the wages of the citizens of the origin country. Furthermore, the labor skills that are available in the origin country from the skills in demand may be different due to the change of the productive structure. Another issue is that sometimes the countries cannot control the migration flow and may be overwhelmed, leading to migration pressures and problems of migration management [79]. Furthermore, more international migrations mean more need for integration into the country's development framework [80], which may cost time and money. Too much migration may pressure public services such as education and healthcare, racial discrimination, and job losses due to the increased competition with migrants [59,81].

We also analyzed the effect of logistic performance on the relationship between economic complexity and human development, as we believe that it is a variable that also has its importance to understanding the phenomenon better. Through an adequate logistics system, resources and other economic values are used and efficiently distributed. Logistics performance is essential for high-income nations as they take part in international vertical specialization [82]. Logistics and transportation connect countries, disseminate technologies and innovations of products and processes, primordial for global value chains. Increase in incomes, the creation of jobs, decrease in income inequalities, women's economic empowerment, and better environmental sustainability, improve with the higher performance of trade that is allowed by transportation and logistics [83]. Today, logistics performance has a meaningful impact on human development, mainly through its strong influence on vaccine distribution. The COVID-19 pandemic is a global health crisis and a human development crisis [84], as it shows the inequalities in terms of human development and healthcare system that are primordial to deal with this pandemic. Vaccines are fragile and need to follow specific regulations to be carried, stored, and disseminated to be efficient. According to Pasadilla and Shepherd (2012), countries with high logistic performance can efficiently distribute the vaccines to the different medical centers in the country [85]. However, logistics performance, and in particular transport infrastructures, are capital intensive fixed assets [86]. Unfortunately, this type of asset is defenseless against misallocations and malinvestments. Moreover, logistics investments may become "wealth consuming" instead of "wealth-producing" if the money is used on projects that do not bring back economic returns. Therefore, in the case of counterproductive projects, and even though the logistics performance is good, a country's economy's resources may be consumed. Nevertheless, national logistics projects are usually funded by public funds, sometimes constrained by special interest groups. As a result, even though the project was first intended to be a stimulant for growth, economic returns may be poor due to lobbying. Finally, large logistics projects, such as public transportation, may be out of budget due to an incompetent or insufficient cost control mechanisms. Countries like the USA often encounter this type of engineering error in their infrastructure projects [86].

### 5.2. Contribution, Limitations, and Future Research

This article contributes to the field of research by analyzing the interrelations between human development and other factors. The results show that the relationship between economic complexity and human development is influenced by two moderators—international migration and logistics performance. Logistics performance seldom has been used as a human development factor; however, we show that policy makers should also focus on this issue. Policy makers should ensure that proper regulations are put in place when it comes to new technologies and how companies produce and disseminate their products and services. The methodology we used to analyze endogeneity could be criticized. To deal with endogeneity, we used lagged variables that are simple to implement and intuitively appealing. However, many authors explained that the best way is to use instrumental variables [87,88]. Nonetheless, finding a promising instrumental variable is not that easy, and researchers should have an extended comprehension of the study's practical context.

Future research would be interesting to add instrument variables to confirm that our model does not have endogenous issues.

Moreover, it would be interesting to analyze the impact of industries' repatriation during the COVID-19 pandemic on logistics and immigration. On the positive side, due to COVID-19, there has been an increase in e-commerce and market demand. However, the pandemic harmed global supply chains [89]. Cargos were blocked at the ports, and there were not enough truck drivers to pick up the containers due to travel regulations. On land, except during the periods of lockdown, transport was partially available. Many companies could not maintain their cash flow because of the pandemic, leading them to different options: total closure, temporary closure, or employee cut down. Therefore, and in many cases, migrants had to move back to their home country [90]. Taiwan Semiconductor Manufacturing Company (TSMC), a Taiwanese company, is an example of repatriation of the country's industry. Even though the labor costs are higher than in China, and the company had logistics costs to repatriate the company, it is beneficial for the country's GDP as it creates more job opportunities for the locals. Furthermore, the government also proposed urbanization projects, which means that the area will develop and increase the local economy and become technology-wise as TSMC is a semi-conductor company.

**Author Contributions:** Conceptualization, E.S.L.C. and F.H.; formal analysis, E.S.L.C.; methodology, E.S.L.C. and F.H.; supervision, F.H.; writing—original draft, E.S.L.C.; writing—review and editing, E.S.L.C. All authors have read and agreed to the published version of the manuscript.

**Funding:** This research received no external funding.

**Institutional Review Board Statement:** Not applicable.

**Informed Consent Statement:** Not applicable.

**Data Availability Statement:** Publicly available datasets were analyzed in this study. This data can be found in Appendix B.

**Conflicts of Interest:** The authors declare no conflict of interest.

## Appendix A. List of Countries

High HDI countries: Albania, Argentina, Australia, Austria, Belarus, Belgium, Bosnia and Herzegovina, Bulgaria, Canada, Chile, Costa Rica, Croatia, Czech Republic, Denmark, Estonia, Finland, France, Germany, Greece, Hong Kong, Hungary, Iran, Ireland, Israel, Italy, Japan, Jordan, Kazakhstan, Kuwait, Latvia, Lebanon, Lithuania, Malaysia, Mexico, Netherlands, New Zealand, Norway, Oman, Panama, Poland, Portugal, Romania, Russia, Saudi Arabia, Serbia, Singapore, Slovenia, South Korea, Spain, Sri Lanka, Sweden, Switzerland, Trinidad and Tobago, Ukraine, United Arab Emirates, United Kingdom, United States, Uruguay, and Venezuela.

Low HDI countries: Algeria, Angola, Azerbaijan, Bangladesh, Bolivia, Botswana, Brazil, Cambodia, Cameroon, China, Colombia, Cote d'Ivoire, Dominican Republic, Ecuador, Egypt, El Salvador, Ethiopia, Gabon, Ghana, Guatemala, Guinea, Honduras, India, Indonesia, Jamaica, Kenya, Laos, Madagascar, Mauritania, Moldova, Mongolia, Morocco, Mozambique, Namibia, Nicaragua, Nigeria, Pakistan, Paraguay, Peru, Philippines, Senegal, South Africa, Sudan, Syria, Tanzania, Thailand, Togo, Tunisia, Turkey, Turkmenistan, Uzbekistan, Vietnam, Yemen, Zambia, and Zimbabwe.

## Appendix B. Variables' Definitions and Sources

| Variables | Definitions and Measurements | Sources |
|---|---|---|
| Human development index (HDI) | Three indicators measure the HDI. First, long and healthy life is assessed by life expectancy at birth. Second, being knowledgeable is based on the mean number of years of schooling for adults aged 25 years and more, and the expected years of schooling for children of school entering age. The last dimension is a decent standard of living, measured by gross national income per capita. Then, the three dimensions' scores are aggregated into a composite index using a geometric mean. | UNDP http://hdr.undp.org/en/content/human-development-index-hdi |
| Economic complexity index (ECI) | "The complexity of an economy is related to the multiplicity of useful knowledge embedded in it. Because individuals are limited in what they know, the only way societies can expand their knowledge base is by facilitating the interaction of individuals in increasingly complex networks in order to make products. We can measure economic complexity by the mix of these products that countries are able to make." | OEC https://oec.world/en/rankings/country/eci/ |
| Logistics performance index (LPI) | This index is based on surveys that are given to professionals in the international logistics field. The respondents are asked to assess markets based on six core dimensions on a scale from 1 (worst) to 5 (best): (1) the efficiency of customs and border management clearance; (2) the quality of trade and transport-related infrastructure; (3) the ease of arranging competitively priced international shipments; (4) the competence and quality of logistics services; (5) the ability to track and trace consignments; and (6) the frequency with which shipments reach consignees within the scheduled or expected delivery time. | World Bank https://lpi.worldbank.org/ |
| International immigration (IM) | International migration is measured by international migrant stock. It is defined as "the number of people born in a country other than that in which they live. It also includes refugees" (World Bank). The data are obtained from population statistics. The World Bank has two ways to obtain the estimates. First, it uses data from the foreign-born population—residents in one country but born in another. If the estimates are unavailable, it will use data from the foreign community—residents in a country but are citizens from another nation. | World Bank https://data.worldbank.org/indicator/SM.POP.TOTL.ZS |
| Gender inequality index (GII) | GII is measured by three dimensions: health, empowerment, and labor market. The health dimension focuses on reproductive health. It is measured by the maternal mortality ratio and adolescent birth rates. Empowerment focuses on the proportion of parliamentary seats occupied by females and the proportion of adult females and males aged 25 years and older with at least some secondary education. Finally, the labor market is assessed by the labor force participation rate of female and male populations aged 15 years and older. There are more disparities between women and men when the gender inequality index reaches high values. | UNDP http://hdr.undp.org/en/composite/GII |
| World governance indicators (WGIs) | They are many ways to measure governance, such as the control of corruption, ethics, integrity, accountability, transparency, and human rights [37]. In this paper, two of the indicators of the world governance indicators (WGIs) are used as they had the most substantial impact in our pilot study. Regulatory quality (RQ) is the "perceptions of the ability of the government to formulate and implement sound policies and regulations that permit and promote private sector development". Rule of law (RL) is the "perceptions of the extent to which agents have confidence in and abide by the rules of society, and in particular the quality of contract enforcement, property rights, the police, and the courts, as well as the likelihood of crime and violence". | World Bank Group https://datacatalog.worldbank.org/dataset/worldwide-governance-indicators |

| Variables | Definitions and Measurements | Sources |
| --- | --- | --- |
| Youth unemployment (WE) | "Share of the labor force ages 15–24 without work but available for and seeking employment". | World Bank https://data.worldbank.org/indicator/SL.UEM.1524.NE.ZS |
| Civic activism (SDCA) | Civic activism is measured by "the access to civic associations, participation in the media, and the means to participate in civic activities such as nonviolent demonstration or petition". | International Institute of Social Studies (University of Rotterdam) https://isd.iss.nl/ |
| Group inclusion (SDGI) | It includes "whether there is systemic bias among members of the community in the allocation of jobs, benefits, and other social and economic resources regarding particular social groups, such as indigenous peoples, migrants, refugees, or lower caste groups". | International Institute of Social Studies (University of Rotterdam) https://isd.iss.nl/ |
| Age dependency ratio, old (SDAO) | "Ratio of older dependents—people older than 64—to the working-age population—those ages 15–64". | World Bank https://data.worldbank.org/indicator/SP.POP.DPND.OL |

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
