# Peer review of "Economic Complexity and Human Development: Moderated by Logistics and International Migration"

_sustainability, doi:10.3390/su13041867_

Round 1

Reviewer 1 Report

My comments are as follows:

1. The paper is interesting but needs stronger motivation. What is the novelty of this paper?

2. Robustness of the results need to be established.

3. How was potential endogeneity investigated and addressed?

4. Policy implications need to be clarified.

5. Link with sustainability needs to be clearly addressed.

Author Response

Dear Mr. or Mrs.,

Thank you for your valuable comments. We did major revisions following your comments and those of the other reviewers.

Regarding the novelty of the paper, we decided to change the mediator and use environmen-tal health instead to avoid too much similarities with a previous article already published. Furthermore, we strongly believe that environment is a primordial variable in accessing sus-tainability of developed countries. We also included two social control variables: youth un-employment and suicide mortality rates. We still provide some comments regarding GINI as a mediator, as it also has strong impact in the model.
To assess the endogeneity of the results we decided to use different years lagged. In future research, we will find an instrumental variable to confirm that we do not have endogenous issues.
We added a short paragraph for the policy implications, and added different parts throughout the paper to link and better explain the relationship to sustainability.

Reviewer 2 Report

Dear authors,

Thank you very much for your efforts writing this paper.

This manuscript studies the economic complexity and sustainability in 28 developed countries from 1990 to 2017 by applying a hierarchical linear modelling. The topic of the paper is interesting and relevant.

Results show the role of international migration and logistics performance. The most relevant strength of this investigation is to demonstrate that the impact of international migration is not as important as logistics performance but is still primordial as both moderators make the economic complexity index significant.

The most relevant weakness of this investigation is not taking into consideration other factors, such as social, cultural, and environmental indicators, that could help to understand sustainable development and its relationship with economic complexity. I motivate authors to continue their research in this field and to take into consideration, in future publications, new factors that could provide a new perspective and knowledge.

The following comments aim to improve the manuscript quality further. There are a few issues that I recommend addressing:

C1. Please improve the explanation of sample selection. Why did you selected those 28 developed countries and why no other countries.

C2. In lines 675-678 you show a decrease of the explanatory power compared to the previous tables. Although it is clear that this decrease does not represent the explanatory power of ECI but the overall indicators included in the model, please add an interpretation and discussion on this issue.

C3. It would be very enriching to include public policy recommendations or practical applications of your research.

I congratulate authors for this interesting investigation and wish them the most success in their research activities.

Thank you very much for your efforts and for your valuable scientific contribution.

Author Response

Dear Mr. or Mrs.,

Thank you for your constructive comments. We did major revisions following your comments and those of the other reviewers.

Regarding the sample selection, we added a paragraph in the methodology to explain it in more detail.

In the statistical part, we also added a paragraph to explain in more details that the decrease of the explanatory power of ECI is not due to its power itself.

Regarding the novelty of the paper, we decided to change the mediator and use environmental health instead to avoid too much similarities with a previous article already published. Furthermore, we strongly believe that environment is a primordial variable in accessing sustainability of developed countries. We also included two social control variables: youth unemployment and suicide mortality rates. We still provide some comments regarding GINI as a mediator, as it also has strong impact in the model.

We also added a paragraph regarding the political implications.

Reviewer 3 Report

The paper is written in quite good manner, in good English, but lack a little bit of enthusiasm for reader. Paper is quite long, it makes 12 pages of an introduction and overview of theoretical background. I recommend authors to shorten a paper if possible (as much as possible). Also, paper really resemble looks like to your preview work on the topic - Economic Complexity and the Mediating Effects of Income Inequality: Reaching Sustainable Development in Developing Countries - even if in present paper you treat developed countries and explanatory variables are different. The added value of present research loses meaning.

I have also some objections about research design:

  1. How have you chosen the selected countries for the research?
  2. For me, the logistics performance is completely incomprehensible explanatory variable among other explanatory variable … with respect to treated phenomena and relationships.
  3. Please, make your results/findings clearer, less technical and clearly legible for readers. In present form, it is really a wee bit confusing and looks like quick course of econometrics. Maybe use some graphs, heatmaps, or find some better way to visualise your findings.
  4. I know that some honest work was done with the paper, but I recommend to shorten it and focus more on results and their presentation.

Author Response

Dear Mr. or Mrs.,

Thank you for your useful comments. We did major revisions following your comments and those of the other reviewers.

Regarding the sample selection, we added a paragraph in the methodology to explain it in more detail.

Regarding the novelty of the paper and the variables used in the model, we decided to change the mediator and use environmental health instead to avoid too much similarities with a previous article already published. Furthermore, we strongly believe that environment is a primordial variable in assessing sustainability of developed countries. We also included two social control variables: youth unemployment and suicide mortality rates. We still provide some comments regarding GINI as a mediator, as it also has strong impact in the model.

A graph has been added at the end of the paper to make the results clearer. And we also deleted some parts of the paper to shorten it.

Round 2

Reviewer 1 Report

The paper has improved.

Author Response

Dear Reviewer,

Thank you very much for all your useful comments.

Following is the second revision of the paper based on yours and the other reviewers comments.

Sincerely
